# Does Decreased Diffusing Capacity of the Lungs for Carbon Monoxide Constitute a Risk of Decompression Sickness in Occupational Divers?

**DOI:** 10.3390/ijerph20156516

**Published:** 2023-08-03

**Authors:** Brice Loddé, Marie-Agnès Giroux-Metges, Hubert Galinat, Hèlène Kerspern, Richard Pougnet, Philippe Saliou, François Guerrero, Pierre Lafère

**Affiliations:** 1ORPHY Laboratory, EA 4324, Western Brittany University (UBO), 29238 Brest, France; 2Occupational Diseases Center, Brest University Hospital, 29609 Brest, France; 3Respiratory Functional Exploration Unit, Brest University Hospital, 29609 Brest, France; 4Department of Biological Hematology, Brest University Hospital, 29609 Brest, France; 5Department of Biochemistry and Pharmaco-Toxicology, Brest University Hospital, 29609 Brest, France; 6ISERM, EFS, UMR 1078, GGB, Infection Control Unit, Western Brittany University (UBO), 29238 Brest, France; 7Environmental, Occupational, Ageing (Integrative) Physiology Laboratory, HE_2_B, 1160 Brussels, Belgium; 8DAN Europe Research Department, 1160 Brussels, Belgium

**Keywords:** decompression sickness, diving, hemostasis, inflammation, renin–angiotensin system, respiratory function test, vascular gas emboli

## Abstract

Long-term alterations of pulmonary function (mainly decreased airway conductance and capacity of the lungs to diffuse carbon monoxide (DLCO)) have been described after hyperbaric exposures. However, whether these alterations convey a higher risk for divers’ safety has never been investigated before. The purpose of the present pilot study was to assess whether decreased DLCO is associated with modifications of the physiological response to diving. In this case–control observational study, 15 “fit-to-dive” occupational divers were split into two groups according to their DLCO measurements compared to references values, either normal (control) or reduced (DLCO group). After a standardized 20 m/40 min dive in a sea water pool, the peak-flow, vascular gas emboli (VGE) grade, micro-circulatory reactivity, inflammatory biomarkers, thrombotic factors, and plasmatic aldosterone concentration were assessed at different times post-dive. Although VGE were recorded in all divers, no cases of decompression sickness (DCS) occurred. Compared to the control, the latency to VGE peak was increased in the DLCO group (60 vs. 30 min) along with a higher maximal VGE grade (*p* < 0.0001). P-selectin was higher in the DLCO group, both pre- and post-dive. The plasmatic aldosterone concentration was significantly decreased in the control group (−30.4 ± 24.6%) but not in the DLCO group. Apart from a state of hypocoagulability in all divers, other measured parameters remained unchanged. Our results suggest that divers with decreased DLCO might have a higher risk of DCS. Further studies are required to confirm these preliminary results.

## 1. Introduction

Decompression sickness (DCS) carries a high potential of morbidity [1]; thus, a better understanding of decompression physiopathology is needed to mitigate the risk of death or irreversible neurological sequelae. Over the last decade, this has slowly improved, but many uncertainties remain [2].

Circulating vascular gas emboli (VGE) are considered the key actors in the development of DCS and markers of decompression stress [3]. Although it cannot be used to predict DCS, a statistical correlation between the VGE grade and the risk of DCS has been established [4]. However, the relationship between inert gas, VGE and the human body is more complicated than the sole application of physical law. It is no longer required to consider VGE an “occluding” object in the blood vessel to explain decompression disorders. Indeed, VGE have been associated with a chain of biological events (including pro-inflammatory, pro-coagulating and vascular changes) which, having elicited an immune response, lead to DCS [5]. Exactly which mechanisms underpin these phenomena is not known. Therefore, defining personal factors influencing VGE production and physiopathological effects will offer the option to have individualized decompression schedules optimized to each diver. Within these individual factors, the role of the lungs has been poorly studied.

Scuba diving exposes the lungs to hyperoxia [6], cold and dry breathing gases [7], blood shift induced by immersion [8] and decompression-induced vascular gas emboli [9]. This leads to increased pulmonary arterial pressure and vascular resistance [10] after diving, and can alter the pulmonary function. Indeed, previous studies have reported an alteration of large and small airway conductance, as assessed by the lower forced expiratory volume in 1 s (FEV1), either as an absolute value or as an expressed percentage of the forced vital capacity (FEV1/FVC) and forced expiratory flow at low volumes (FEF50 and FEF 75) [11,12,13]. They also reported an acute but transient decrease in the capacity of the lungs to diffuse carbon monoxide (DLCO) after shallow dives [13] and a gradual development of chronic reduced DLCO with repeated hyperbaric exposures [11,12]. If scuba diving can alter DLCO in an acute or chronic way, it seems logical to question the opposite relationship. To the best of our knowledge, no investigation has ever considered whether decreased DLCO could have an influence on decompression physiology.

Yet, the lungs are essential to the physiopathology of decompression. They have the capacity to diffuse, conduct and transfer the gases to and from the bloodstream [14], thus playing a major role in eliminating VGE [15]. Furthermore, besides their role in gas exchanges, the lungs are related to the vascular system via synthetic, metabolic, and regulating functions that enable the body to adapt to environmental conditions [16]. Firstly, it is widely believed that most of the conversion of angiotensin I to angiotensin II takes place in the lungs [17], giving the lungs the potential to influence vascular dysfunction seen in the decompression process. Secondly, the pulmonary endothelium has anticoagulant properties. A strong expression of thrombomodulin occurs on the surface of the endothelial cells [18], which react with thrombin, resulting in the inactivation of coagulation factors Va and VIIIa. The pulmonary endothelium also prevents primary hemostasis by producing platelet aggregation inhibitors, which, in turn, might counteract the pro-thrombotic components associated with diving. However, when damaged, the pulmonary endothelium becomes particularly thrombogenic [19]. Finally, the respiratory mucosa has a cellular and support network that is actively involved in the immune and/or inflammatory response (lymphocytes, complement proteins, etc.) [16].

Therefore, the purpose of this pilot study is to answer whether reduced, yet not pathological, DLCO conveys a higher risk of DCS. Our hypothesis was that it could interfere either with the capacity of inert gas exchanges during hyperbaric exposure (and, therefore, the amount of circulating VGE) or with the physiological pathways (namely, endothelial function, coagulation, and inflammation) involved in the physiopathology of the decompression.

## 2. Materials and Methods

This observational cohort study was approved by the Ethics Committee at Brest Teaching Hospital (CPP Ouest N°6-CPP 916), and all the procedures were conducted in accordance with the Declaration of Helsinki. Prior to enrollment, informed written consent was obtained from the participating divers. This manuscript adheres to the applicable STROBE guidelines.

### 2.1. Study Participants

Divers were selected from a large population of occupational divers whose ongoing medical monitoring was carried out at the Occupational Disease Center at Brest Teaching Hospital. Participants had to be of lawful age (>18 years old), in possession of a valid French professional diving certification, either class I (maximum authorized depth of 30 m on air) or class II (maximum authorized depth of 50 m on air), and continuously active for at least 2 years with a valid medical certificate.

The initial screening enabled the collection of the participants’ past medical history, lifestyle habits (smoking habits, physical activity, medication intake) and demographics (height, weight, Body Mass Index (BMI), percentage of fat mass using bio-impedancemetry (BF306, Omron Healthcare, Hoofddorp, the Netherlands), blood pressure, heart rate, tympanic membrane appearance and peak expiratory flow (PEF).

Prior to inclusion, each participant underwent repeated lung function testing (LFTs) by a trained technician (Jaeger Master Screen Body, CareFusion, Hochberg, Germany): one year before the study for selection, at inclusion for confirmation of group allocation, and one year after the study for follow-up. The forced vital capacity (FVC), forced expiratory volume in 1 sec (FEV1), forced maximal mid-expiratory flow (FEF25–75) and DLCO were recorded for each participant, as well as the total lung capacity (TLC) and slow vital capacity (SVC). The alveolar volume (VA) was calculated according to an accepted method [20]. The results were expressed as a fixed number, percentages, and the standard deviation of the expected values. According to the global lung function initiative [21], we added the Z-score from the mean.

Based on DLCO measurement, VA and kCO factor (Krogh’s kCO = DLCO/VA), the divers were allocated either to the DLCO group (*n* = 7) if the DLCO and kCO values were above one standard deviation of the expected value (<85% of the expected value), which corresponds to a Z-score < −0.80 for DLCO and <−1.15 for kCO, or to the control group (*n* = 9) if the DLCO and kCO values were within one standard deviation of the expected value (≥85% of the expected values, with Z-scores > −0.45 for DLCO and >−0.85 for kCO).

### 2.2. Exposure and Measurements

Participants were instructed not to dive 72 h prior to the experimental dive and were randomly assigned to a specific time slot either on 29 March or 30 March 2016. On the diving day, they were assessed and confirmed to be fit to dive by a qualified diving medicine specialist and by an occupational physician.

Then, each diver performed an air dive at a depth of 20 msw (0.3 Mpa) for 40 min in a pool located at the National Institute for Ocean Science (IFREMER). Since this depth-time profile falls within decompression limits according to the French Ministry of Labor, a mandatory 3 min safety stop at 3 m (0.13 MPa) was included in the profile. The descent speed was 15 msw/min (150 kPa/min) and the ascent speed to the surface was 10 msw/min (100 kPa/min). The water temperature was 11 °C. Therefore, subjects each wore a 5–7 mm neoprene wetsuit that needed to be in good condition. Finally, the subjects were asked to swim slowly to maintain a low- to moderate-intensity energy expenditure related to exercise, i.e., swimming in a pre-defined circle at the bottom of the pool.

The peak expiratory flow (PEF) was measured pre-dive and 25 min post-dive in a standing position with the nose pinched closed (Mini-wright, Mediflux, Croissy Beaubourg, France). Each participant made three attempts, and the best results were recorded for further analysis.

After the dive, precordial VGE signals were measured using a portable echocardiography device (Sonosite Edge, FUJIFILM Sonosite Inc, Amsterdam, the Netherlands) with a sectorial array ultrasound 5-1MHz probe (P21v) by an experienced researcher (PL) who was blinded to the group allocation. Measurements were carried out in the left lateral supine position. A cardiac four-chamber view was obtained by placing the probe at the level of the left fifth intercostal space. It was necessary to modify the standard four-chamber view by rotating the probe slightly ventrally so the right atrium and ventricle can be fully visualized. Each diver was investigated at two time points: 30 min and 60 min after surfacing. These investigations were carried out at the end of a period during which subjects remained at rest and following active provocation by three deep knee bends.

The echocardiographic VGE signals over a 1 min recording were assessed and scored according to the Eftedal–Brubakk categorical score [22]. However, due to the limited number of subjects, the simplified Bubble Grading System (BGS) was also applied [23].

The cutaneous microvascular function was assessed in the supine position, 30 min after surfacing via laser Doppler flowmetry (LDF) (Periflux PF 5001, Perimed, Crapone France) coupled with iontophoresis, as previously described by Lambrechts et al. [24,25]. In brief, endothelium-dependent vasodilation is induced with a 1% acetylcholine (ACh) chloride solution administered with an anodal current (35 s, 0.10 mA), while a cathodal current (35 s, 0.10 mA) is used to deliver a 1% sodium nitroprusside (SNP) solution to assess the endothelium-independent vasoreactivity. Then, responses to ACh and SNP are presented as the percentage of cutaneous vascular conductance (CVC) variation from the baseline after iontophoretic stimulation, while the CVC is calculated as the ratio between the cutaneous blood flow and the mean arterial pressure.

Blood samples were drawn before the dive and 90 min after surfacing (i.e., after all the other measurements had been taken) from a clean median cubital vein in the antecubital fossa using 4.5 mL tubes (Vacutainer^®^ (CTAD), Becton Dickinson, Franklin Lakes, NJ, USA). A complete blood count (including a platelet count) was obtained (XE 5000 automat, Sysmex, Kobe, Japan). Other samples were centrifuged on-site at 3000× *g* and 4 °C for 10 min. Then, the plasma was aliquoted and stored at −80 °C until the assay. The plasma concentration of endothelial microparticles (Hyphen^®^ test), P-selectin, fibrin monomers, plasma aldosterone concentration, plasma activity of angiotensin-converting enzyme (ACE) and the activated complement component 5 protein (C5a), as well as the plasma concentrations of interleukin 6 (IL-6) and 10 (IL-10), were measured. The thrombin generation test (TGT) was measured on a Fluoroscan Ascent fluorometer (Thermo Labsystems, Helsinki, Finland) with a commercially available kit (STG-Bleedscreen, Stago, Asnières, France) and with two different reagents consisting of a mixture of phospholipids (4 µM) and tissue factor (1 pM for Platelet-Poor Plasma (PPP)-Reagent LOW and 20 pM for PPP-Reagent HIGH). The TGT values are expressed as a percentage of the values of normal pool plasma [26].

### 2.3. Statistical Analysis

Statistical analyses were performed using Prism version 9.2 for MacOS (Graphpad Software, La Jolla, CA, USA). Dichotomous data are presented as proportions, while continuous data are presented as mean ± standard deviation (SD). Because of the small sample size, only non-parametric tests were used. Differences between groups in continuous variables were assessed with the Mann–Whitney U test, while the within-group comparison was carried out using a Wilcoxon test. Differences in categorical variables were analyzed using the chi-squared test. A *p*-value < 0.05 was considered significant for all statistical tests.

## 3. Results

### 3.1. Baseline Characteristics

Because of an illness two days before the study dive, only 15 out of the 16 volunteers were included in this analysis. As regards age, weight, BMI, percentage of body fat, gender (one woman in each group), smoking habits, diving experience, number of dives/year and mean depth of occupational diving, the groups were similar. The main difference between the groups corresponds to the allocation criteria, which were the DLCO and DLCO/VA expressed either in mL/min/mmHg, as a percentage of the theoretical value, as a standard deviation, or as a Z-score. Anecdotally, there is an isolated significant difference in height between the two groups (*p* = 0.02, Mann–Whitney) (Table 1).

### 3.2. Peak Expiratory Flow

Twenty-five minutes post-dive, the PEF decreased by 3.4 ± 5.8% in all divers from 637 ± 99 to 612 ± 83 L/min. However, this difference was not statistically significant in either the within-group comparison (control group: *p* = 0.09; DLCO group: *p* = 0.81, Wilcoxon) or in the between-group comparison (pre-dive: *p* = 0.41; post-dive: *p* = 0.93, Mann–Whitney).

### 3.3. Vascular Gas Emboli

Although VGE were recorded in all divers, no cases of DCS occurred. Nonetheless, independently of the measure either at rest or after provocation, at 30 or 60 min, the maximum VGE grade (Figure 1) was significantly higher among divers with a reduced DLCO (*p* < 0.0001, chi-squared test). Because of the low number of divers for each bubble grade, to analyze the kinetic of VGE, we chose to apply the BGS scoring system [23], which allowed us to split the scale into a low grade (VGE grade < 3) and a high grade (VGE grade ≥ 3) and to consider grade 0 separately. Our data suggest that VGE kinetics differ between the DLCO group and the control group (Figure 2). Compared to the control group, the DLCO group exhibits a significantly delayed VGE peak with a higher VGE grade 60 min post-dive both at rest (*p* = 0.008, chi-squared) and after a provocative maneuver (*p* < 0.0001, chi-squared). At 30 min, an almost significant difference was only found for the at-rest measurement (*p* = 0.05, chi-squared).

### 3.4. Hemodynamics and Microvascular Function

The macro- and microvascular functions were not altered by the experimental dive. Indeed, neither the divers’ heart rate and mean blood pressure, nor the CVC response to ACh (control group: 435 [370–629]% vs. 512 [397–628]%, *p* = 0.952; DLCO group: 408 [282–417]% vs. 412 [224–554]%; *p* = 0.916, Wilcoxon), or the responses to SNP (control group: 428 [276–489]% vs. 433 [223–783]%, *p* = 0.952; DLCO Group: 484 [336–761]% vs. 351 [241–536]%, *p* = 0.173, Wilcoxon) were significantly modified. No between-group significant difference was identified either (*p* > 0.05, Mann–Whitney) (data not shown).

### 3.5. Blood Analysis

The hemoglobin level (14.7 ± 1.1 g/dL vs. 14.8 ± 1.3 g/dL, *p* = 0.79, Wilcoxon), hematocrit (43.3 ± 3.1 vs. 44.0 ± 2.9%, *p* = 0.06, Wilcoxon), and platelets (260 ± 48 10^3^/mm^3^ vs. 270 ± 55 10^3^/mm^3^, *p* = 0.13, Wilcoxon) were not significantly affected by the dive, as opposed to the plasma volume variation calculated according to the Dill and Costill equation [27], which was significantly decreased by 1.7 ± 1.5% (*p* = 0.0004, Wilcoxon signed ranked test). No between-group difference was identified (*p* > 0.05, Mann–Whitney).

Changes in coagulation-related and platelet activation biomarkers are summarized in Table 2 and Table 3.

The concentration of endothelial microparticles, P-selectin and fibrin monomer were not modified by the experimental dive. Of note, there was a significant between-group difference for P-selectin, but this difference was significant both pre-dive (*p* = 0.046, Mann–Whitney) and post-dive (*p* = 0.047, Mann–Whitney). There was also an almost significant increase in the fibrin monomer production in the DLCO group (3.26 ± 1.5 vs. 21.1 ± 42.58 µg/mL, *p* = 0.05, Wilcoxon). However, we question the validity of these data as they were strongly influenced by one outlier within a group of six people.

The TGT results were significantly affected by the reagent used. With the PPP-reagent LOW, which is more sensitive for exploring the extrinsic coagulation pathway due to a low tissue factor concentration [28], the results indicated a state of hypocoagulability (Table 3), while with PPP-reagent HIGH, no changes were observed (data not shown). The prolonged lag-time reduced area under the curve (ETP), low peak height, decreased velocity, decreased velocity index, and increased time to peak after the dive were similar in both groups.

Inflammatory biomarkers such as IL-6 (3.9 ± 7.7 pg/mL vs. 3.1 ± 3.3 pg/mL, *p* = 0.5, Wilcoxon), IL-10 (2.7 ± 6.8 pg/mL vs. 3.1 ± 7.3 pg/mL, *p* = 0.28, Wilcoxon), and activated complement component C5a (528.3 ± 225.3 ng/mL vs. 548.1 ± 267.2 ng/mL, *p* > 0.99, Wilcoxon) were not modified irrespective of the group.

Although the ACE concentration did not change (Figure 3), the plasma aldosterone concentration was reduced by 30.4 ± 24.6% in the control group (182.4 ± 89.7 pmol/L vs. 108.8 ± 15.8 pmol/L, *p* = 0.015, Wilcoxon) and by 1.8 ± 6.8% in the DLCO group (196.8 ± 114.7 pmol/L vs. 193.3 ± 106.9 pmol/L). However, this latter result was not significant (*p* = 0.81, Wilcoxon). Post-dive, there is a significant between-group difference (*p* = 0.03, Mann–Whitney).

## 4. Discussion

To the best of our knowledge, this is the first study to investigate the impact of pulmonary function on decompression physiology. Although most of our results are not related to the decreased DLCO, some of them raise questions about the importance of lung function in diving safety.

Control and DLCO groups showed similar pulmonary function except for DLCO. Indeed, pulmonary volumes and flows were not different between the two groups. Nevertheless, a reduction in ventilatory flow has been reported repeatedly after wet scuba diving [6]. This reduction appears to be more marked in divers with impaired ventilatory function, as indicated by a greater decrease in PEF in divers with a history of asthma compared with control divers [24]. Consistent with existing data in the literature, we found that PEF was lower after the dive, with a magnitude similar to that previously reported [24]. However, the decrease was not different between the two groups.

It is acknowledged that DCS is a multifactorial pathology. Although the exact sequence of events leading to DCS is not yet fully elucidated, its physiopathology includes inflammation [29], coagulation [30] and vascular dysfunction [31].

Impairment of the vascular reactivity both in large conductance arteries and in microcirculation has been reported after diving [24,32]. In the cutaneous microcirculation, the effect of diving is independent of the presence of VGE [24] and is mainly due to the alteration of endothelium-independent vasorelaxation [32]. In the present study, both endothelium-dependent and -independent reactivity remained unchanged after diving regardless of the group. Similarly, post-dive inflammatory reaction is classically reported. In addition, its intensity is greater in the presence of DCS, as indicated by the increase in the pro-inflammatory proteins IL-1β and IL-6 [33] and the decrease in the inflammatory protein IL-10 [34], which are greater in the presence of DCS. Two hypotheses might explain the lack of changes in both microvascular reactivity and inflammation biomarkers in the present study. First, most of the previous literature on inflammation biomarkers or platelet count were obtained from experimental dives deeper than 30 m [25,35]. It is then possible that our dive profile (20 m) was not provocative enough to induce an inflammatory response or platelet activation. Second, it has been suggested that depth may have an influence on the biological response [36]. Therefore, it is possible that these phenomena occurred but with a reduced amplitude and/or a delayed onset not detected via blood sampling taken too soon after the end of the dive (1.5 h) [37].

In addition to its involvement in inflammation, the complement system also enhances coagulation, making it a link between these two functions [38]. A relationship between complement activation and susceptibility to DCS was also reported. However, we did not find any differences in the plasmatic concentration of C5a, the final component of the alternative pathway, by diving between groups. This is coherent with the lack of modifications of IL-6 and IL-10, and with previous studies that showed that C5a is usually not increased with diving [39].

Finally, scuba diving has been associated with a state of hypercoagulability [40] characterized by the activation and aggregation of platelets and decreased concentrations of fibrinogen, and factor X and XII, although with a possible activation of fibrinolysis [41]. Of note, compared to asymptomatic divers, thrombotic factors are 2.4- to 11.7-times higher in cases of DCS [42]. Nonetheless, irrespective of the group allocation, we report the opposite, i.e., a state of hypocoagulability. Since TGT gives comprehensive insights into the function of both pro- and anticoagulation drivers [28], it is possible that antithrombotic factors were secreted in response to this dive. An increase in tissue factor pathway inhibitor (TFPI) could explain this state of hypocoagulability diagnosed with the reagent LOW but not the reagent HIGH. Indeed, in the latter case, the high tissue factor concentration possibly exceeded the neutralizing capacities of TFPI. However, this hypothesis is supported by an increased TFPI plasmatic concentration in the case of endothelial dysfunction [43], a feature commonly seen after scuba diving [24]; it was also demonstrated that dry hyperbaric exposure did not induce any change in plasmatic TFPI [41]. Alternatively, it is possible that the results have been influenced by increased fibrinolysis associated with a decreased level of α2-antiplasmin among divers [44]. These specific results illustrate the lack of information regarding the interactions between the different coagulation pathways, clotting cascades, and platelets, requiring further research. Regarding thrombotic factors, it is also interesting to mention that P-selectin, an adhesion molecule that initiates the attachment of leukocytes circulating in the blood to platelets and endothelial cells, at sites of tissue injury and inflammation, exhibited a significantly higher concentration in the DLCO group compared to the control group both pre- and post-dive. Since DLCO mostly reflects the red cell component of the overall diffusing conductance [45], we may hypothesize that reduced DLCO is associated with higher vascular pressure in the lung capillaries compared to the control group. Should it be the case, then even a modest pressure challenge in the lung venular capillaries could induce mitochondrial responses by the endothelial cells and the initiation of P-selectin expression [46]. However, confirming this hypothesis was outside the scope of this work. Nonetheless, because of its properties, independently of the underlying physiopathological mechanism, a higher concentration of P-selectin observed in the DLCO group might be considered an individual risk factor for DCS.

Compared to the control group, reduced DLCO also impacted key factors for DCS, such as the VGE grade and kinetics. Indeed, we found that latency to VGE peak was significantly increased in the DLCO group (60 vs. 30 min in the control group) along with a significantly higher maximal VGE grade (VGE grade ≥ 4, 17% vs. 11%). This is relevant to the hypothesis establishing the elimination time-lapse of VGE. When VGE interface with the alveolar–capillary barrier, they diffuse their gas contents into the alveoli as per Fick’s law [47]. When DLCO is reduced, this means that the VGE elimination capacity is also reduced, which could account for a belated peak in VGE. We assumed that altered DLCO could slow nitrogen removal and interfere with VGE elimination. However, diffusion is a two-way phenomenon. Movement across selectively permeable membranes occurs from an area of high concentration to an area of low concentration. Consequently, DLCO could also reduce the amount of nitrogen diffusing into the tissues during body saturation. Although significant inter-personal variability was demonstrated despite all divers following an identical protocol in controlled pool conditions [48], the higher VGE grade in the DLCO group advocates against this hypothesis. Indeed, this latter observation supports the hypothesis of supersaturation. In fact, when the differential between intravascular tissue tension and the alveolar blood pressure is larger, there is a greater risk of VGE formation [49]. Despite the absence of any significant change in platelet count, which is usually correlated with the amount of VGE [50], it is preferable to consider changes in DCLO as a potential pathogenic risk linked to decompression. Alternatively, it could be the case that in subjects with impaired DLCO, nitrogen removal via the pulmonary filter is slower and therefore delayed in time. Further studies with measurements performed until the 90th minute would be useful to assess this hypothesis.

Finally, the most conclusive result of this work is the significant decrease in aldosterone plasma concentration in the control group (−30.4 ± 24.6%) versus none in the DLCO group. Although the plasma ACE levels may increase after certain dives and reach significant peaks after DCS occurred [51], the plasma activity of ACE was not altered in our study, which raises the question of what the underlying mechanism is. This question is further raised by the post-dive 2.2% decrease in plasma volume that we found in all 15 divers independently of the DLCO, which is of the same magnitude as already published in the literature [3]. Contrary to our finding, this should have logically been associated with an increase rather than a decrease in the antidiuretic aldosterone. No human study explored this specific aspect. However, a decrease in plasma concentration of angiotensin II (AgII) after simulated dives was reported in asymptomatic rats but not in animals with symptoms of DCS [31]. Among animals, this was explained using specific antagonists. The absorption of an ACE inhibitor decreases the risk of DCS, while the use of AgII receptor type 1 antagonists had no effect. Thus, according to those authors, maintaining a low level of plasma AgII would provide protection against DCS in rats via a low level of aldosterone [52]. Unfortunately, we did not measure the plasmatic AgII level, which does not allow us to confirm this hypothesis. Nevertheless, the hypothesis that the decrease in aldosterone concentration is due to a decrease in AgII concentration during diving is plausible. Taken together, this suggests that a modification of the renin–angiotensin–aldosterone system (RAAS) might be involved in DCS, although it is unclear whether the relationship between the RAAS and DCS is based on AgII, aldosterone or another intermediate compound. In this hypothesis, the abolition of the diving-related protective hypoaldosteronemia in the DLCO groups could therefore also be considered an individual risk factor for DCS. Nonetheless, further studies are needed to determine the role of the RAAS in humans and its importance in the cascade that leads to DCS.

## 5. Limitations

The main limitation of the present study is obviously the very small number of participants. This makes the present study a pilot study and precludes any definitive conclusion. Another limitation might come from the dive profile itself because some of the modifications classically reported post-dive were not found in the present study. Indeed, a decreased response of cutaneous microcirculation to NO donor was not present in our subjects, nor was inflammation. This might indicate that our diving protocol was not provocative enough, although it still induced some high bubble grades. Therefore, it is possible that some pressure-induced effects of diving have been absent, or at least too weak to stimulate inflammation or microvascular alterations. However, this study is the first to have assessed the impact of impaired DLCO on susceptibility to DCS, and we had no previous data that could help us to design a diving protocol that was both safe and sufficiently provocative. The same is true for VGE measurement. Available published data indicate that the peak of VGE occurs around 30–45 min post-dive [53], which led us to measure VGE at 30- and 60 min post-dive in our study. Our present data suggest, however, that the VGE peak may occur later than 60 min. Future studies should address this question by assessing the VGE kinetics more precisely and for a longer period after diving.

## 6. Conclusions

Although we need to be cautious because of the limited number of subjects, this work does offer several advances, while still calling for more thorough investigations. First, even shallow diving can induce VGE with a sufficient duration. Second, compared to a control group, we observed that a decreased, yet not pathological, DLCO is associated with a higher level of P-selectin, a higher VGE grade with delayed kinetics, and, finally, a modified response of the renin–angiotensin–aldosterone system. Considering these findings and the fact that DLCO reflects the functional alteration of the capillary–alveolar membrane, which is thought to play a central role in decompression physiology [14,15], we may hypothesize a lower threshold for DCS. However, it is still too soon to conclude that a decreased DLCO leads to a formal risk of developing DCS, whose prognostic significance still needs to be determined. Nonetheless, it may also have major implications in the long-term medical follow-up of occupational divers, and a definition of acceptable risk in terms of labor protection needs to be specified.

## Figures and Tables

**Figure 1 ijerph-20-06516-f001:**
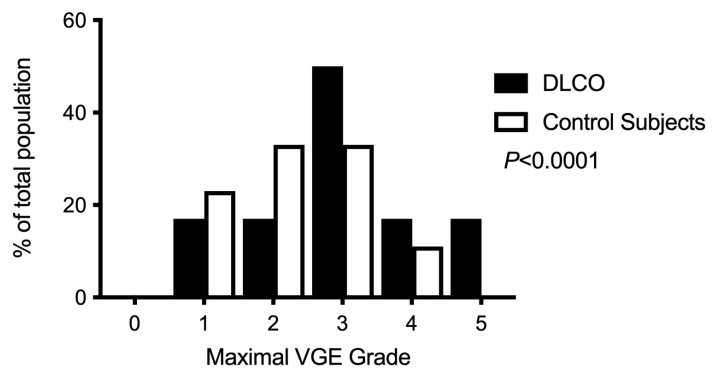
Maximal VGE grade distribution according to the Eftedal–Brubakk categorical score across the experimental population (*n* = 15).

**Figure 2 ijerph-20-06516-f002:**
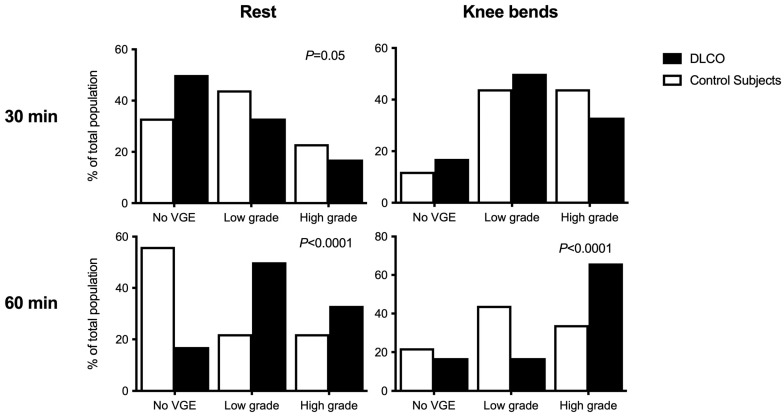
VGE grades distribution according to the BGS scoring system at rest and after three deep knee bends for the control group (*n* = 9) and the DLCO group (*n* = 6). BGS: Simplified Bubble Grade scoring system. Low grade: Eftedal–Brubakk categorial grade < 3, High grade: Eftedal–Brubakk categorial grade ≥ 3.

**Figure 3 ijerph-20-06516-f003:**
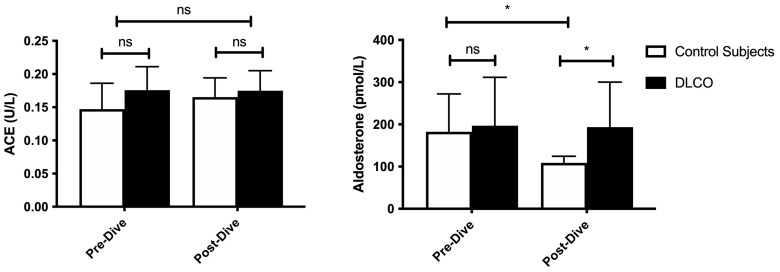
Pre- vs. post-dive angiotensin-converting enzyme (ACE) and aldosterone plasma concentration (mean ± SD). ns: not significant; * *p* < 0.05 for Mann–Whitney (between groups) or Wilcoxon (within groups).

**Table 1 ijerph-20-06516-t001:** Demographics and baseline of both groups, DLCO (*n* = 9) and controls (*n* = 6) (mean ± SD). * *p* < 0.05, ** *p* < 0.01, and *** *p* < 0.001 for Mann–Whitney.

	Control Group (*n* = 9)	DLCO Group (*n* = 6)	*p*
Sex (Males/Females)	8/1	5/1	
Age (years)	41.9 ± 8.1	40.7 ± 7.4	0.77
Height (cm)	180 ± 6.4	172 ± 4.4	0.02 *
Weight (kg)	76.7 ± 9.2	70.2 ± 9.2	0.20
BMI (kg/m^2^)	23.6 ± 1.7	23.3 ± 2.7	0.84
Body fat (%)	18.9 ± 6.6	20.5 ± 3.9	0.6
TLC (L)	7.97 ± 0.36	7.79 ± 0.68	0.812
TLC (%Pred)	114 ± 2.78	119 ± 6.08	0.966
SVC (L)	5.54 ± 0.25	5.34 ± 0.50	0.702
SVC (%Pred)	115 ± 3.72	119 ± 7.45	0.908
FVC (L)	5.77 ± 0.28	5.37 ± 0.37	0.417
FVC (%Pred)	109 ± 3.2	114 ± 5.6	0.500
FEV1 (L)	4.31 ± 0.23	4.14 ± 0.23	0.606
FEV1 (%Pred)	102 ± 2.6	109 ± 4.8	0.223
FEV1/FVC (%)	75 ± 1.9	77 ± 2.8	0.496
FEV1/FVC (%Pred)	93 ± 2.3	95 ± 3.0	0.596
PEF 25–75 (L.s^−1^)	3.49 ± 0.40	3.63 ± 0.39	0.805
PEF 25–75 (%Pred)	85 ± 7.0	98 ± 12.7	0.390
DLCO (mL/min/mmHg)	10.8 ± 1.6	7.8 ± 1.2	0.002 **
DLCO (%)	97 ± 8.1	76.5 ± 7.4	0.0003 ***
DLCO (SD)	0.5 ± 0.4	2.3 ± 0.8	<0.0001 ***
DLCO (Z-score)	0.16 ± 0.5	−1.25 ± 0.6	0.0004 ***
DLCO/V_A_ (Z-score)	0 ± 0.8	−1.61 ± 0.6	0.002 **
Occupational diving experience (years)	11 ± 7.5	15 ± 6.5	0.38
Number of dives/years	55 ± 23	88 ± 68	0.48
Mean depth of occupational diving (m)	15 ± 6	20 ± 11	0.28

DLCO: diffusing capacity of the lungs for carbon monoxide.

**Table 2 ijerph-20-06516-t002:** Pre- vs. post-dive P-selectin, fibrin monomer and endothelial microparticles (MPs) (mean ± SD). * *p* < 0.05 for Mann–Whitney (between groups) or Wilcoxon (within groups).

	Control Group (*n* = 9)			DLCO Group (*n* = 6)			Between Groups	
	Pre-Dive	Post-Dive	*p*	Pre-Dive	Post-Dive	*p*	Pre-Dive	Post-Dive
P-selectin (ng/mL)	0.1 ± 0.01	0.1 ± 0.02	0.64	0.13 ± 0.03	0.14 ± 0.03	0.67	0.046 *	0.047 *
Fibrin monomers (μg/mL)	4.5 ± 1.1	4.75 ± 0.8	0.53	3.3 ± 1.5	21.1 ± 42.6	0.05	0.077	0.113
Endothelial MPs (nM)	0.29 ± 0.25	0.33 ± 0.11	0.12	0.42 ± 0.24	0.47 ± 0.19	0.71	0.144	0.092

**Table 3 ijerph-20-06516-t003:** Pre- vs. post-dive thrombin generation test (TGT) with PPP-reagent LOW. Values are expressed as a percentage of normal pool plasma value from a general population (mean ± SD). * *p* < 0.05, and ** *p* < 0.01 for Mann–Whitney (between groups) or Wilcoxon (within groups).

	Control Group (*n* = 9)			DLCO Group (*n* = 6)			Between Groups	
	Pre-Dive	Post-Dive	*p*	Pre-Dive	Post-Dive	*p*	Pre-Dive	Post-Dive
Lag-time	108.3 ± 7.8	111.6 ± 9.0	0.65	106.2 ± 15.3	119.0 ± 12.9	0.04 *	0.72	0.20
ETP	86.0 ± 11.6	74.9 ± 9.3	0.03 *	85.3 ± 14.3	76.7 ± 13.8	0.38	0.92	0.79
Peak height	110.8 ± 30.4	86.0 ± 16.5	0.04 *	110.8 ± 22.6	92.5 ± 25.4	0.21	0.66	0.71
Time to peak	94.2 ± 9.4	101.9 ± 8.8	0.004 **	90.7 ± 11.1	103.2 ± 10.5	0.03 *	0.43	0.93
Velocity Index	145.3 ± 80.0	95.9 ± 26.5	0.03 *	176.0 ± 88.1	105.4 ± 28.6	0.03 *	0.26	0.62
Velocity	151.9 ± 109.2	86.67 ± 31.8	0.02 *	166.7 ± 83.3	91.3 ± 38.5	0.03 *	0.54	0.88

## Data Availability

The datasets used and analyzed during the current study are available from the corresponding author upon reasonable request.

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
