# Peer review of "Does Decreased Diffusing Capacity of the Lungs for Carbon Monoxide Constitute a Risk of Decompression Sickness in Occupational Divers?"

_ijerph, 2023, doi:10.3390/ijerph20156516_

Round 1
Reviewer 1 Report
Comments to the Author
The authors provide a strong rationale for the importance of studying the role of the lungs in decompression physiology and aim to investigate whether reduced DLCO interferes with the capacity of inert gas exchanges during hyperbaric exposure, possibly altering the physiopathology of decompression. This paper is well-organized, pertinent, and may add to the literature. However, there are several points that require further clarity;
1- Page 2, Line 90: What is the gender of the participants?
2- Page 3, Lines 101-111: You have a conclusion about PEF, but I couldn't see it in your tables or graphs. Also why only the PEF value? If we are talking about pulmonary functions, it is also necessary to present some basic values such as FEV1, FVC, SVC, TLC or FEV1/FVC, which is even more important.
Please provide a new table noting the pre and post-values for these parameters in both groups. Also, if possible, provide predicted values for all pulmonary functions in the same table.
If you cannot provide these data for pre-post values, provide baseline-measurements for both groups in the baseline table.
GENERAL COMMENTS:
1. The manuscript requires language improvement.
2. The topic is important but especially the introduction and discussion sections should be improved significantly. Literature review is nonadequacy.
3. Abstract should be re-edited after changes made in the article.
Reviewer 2 Report
Scuba diving remains a risky activity, particularly because decompression illnesses are relatively unpredictable. In this context, any study allowing to identify subjects with an increased risk of occurrence of a decompression sickness is welcome. This is the case of this study.
The purpose of this study is to investigate, in volunteer professional divers, the effect of diving to 20 meters on the presence or absence of bubbles causing decompression sickness (DCS).
The secondary objective is to see if divers with a reduced (but not pathological) DLCO value have a higher risk of DCS.
The third objective is to measure after a saturating dive whether microcirculatory reactivity, inflammatory biomarkers, thrombotic factors, and plasma aldosterone concentration are greater in divers with higher DLCO values.
This study shows that divers with impaired DLCO have significantly more circulating vascular gas emboli (VGE) after saturation diving.
Although the number of subjects included is small, this study has the merit of highlighting the relationship that may exist in impaired DLCO and the risk of DCS. In this context, the publication of these results should be encouraged.
Furthermore this study also shows that in the group of subjects with impaired DLCO, the level of P-selectin is higher before and after diving. In this group, the aldosterone level is higher at the end of the dive.
Inclusion des sujets :
Dans le groupe DLCO, les valeurs, bien que non pathologiques, sont inférieures de 25 % au valeurs théoriques attendues which correspond to a Z-score <-0.80 for DLCO and <-1.15 for kCO. Cela pose plusieurs questions :
· In the DLCO group, subjects had a 25% alteration in DLCO. It would be interesting to understand the cause of this 25% alteration in DLCO (treatment? toxic exposure, other lung diseases cured but having left sequelae?). In particular, we would have liked to see lung imaging (RMN, TMD) to understand these decreases in DLCO.
· Accordingly, we wonder if it would not be possible to define more precisely the objective of the study. Is it indeed to analyze the normal dispersion of DLCO values within the population of divers free of pulmonary pathology? Are we sure that they do not have a chronic subclinical pathology? How can we explain these alterations in DLCO?
· We understand that the authors hypothesize that the decrease in DLCO is responsible for a reduction in the rate of bubble and nitrogen removal during the decompression phase. But if this is the case, shouldn't the decrease in DLCO also reduce the amount of nitrogen diffusing into the tissues during body saturation? In short, could altering DLCO also reduce VGE?
Results section.
To compare the 60-minute VGE values between the two groups, the authors use a Chi-square test, whereas these are repeated values. Instead, we suggest using a test for repeated measures (Kruskal Wallis test with post hoc test).
Discussion section
In order to make the discussion of this article more robust, it would seem interesting to justify the authors' choice of biological markers for ADD. What guided their choice of biomarkers? We would like to understand these choices because, with the exception of aldosterone, no differences were observed between the two groups.
The authors chose to measure certain biological values because they hypothesized that they were related to either VGE values, DLCO values, or both. In this framework, we would like to know why the authors did not analyze possible statistical correlations between the measured biological values and the VGE and DLCO values, respectively.
In this context, we think it would be helpful if the authors discussed a little more about the decrease in aldosterone in the control group, which is presented as a protective factor for DCS. On the other hand, we wish to discuss hypotheses that would explain why, in the DLCO group, aldosterone values are normal.
The authors cite the work of Tezlaff et al. and Oste et al. who observed an acute but transient decrease in the ability of the lungs to diffuse carbon monoxide (DLCO) after shallow dives. We would like to know why the authors did not perform a DLCO measurement after the dive. This would have allowed us to see how DLCO values varied in the two groups of divers studied.
Section limitations
We think it would be helpful if the authors added a "limitation" paragraph to acknowledge that this is a pilot study whose results have yet to be confirmed. Indeed, although the authors state this, the number of participants is very small, making it difficult to draw conclusions.
Finally, we question whether the kinetics of VGE measurement at the end of the dive are sufficiently long. Usually VGE measurements are performed until the 90th min. It could be that, in subjects with impaired DLCO, nitrogen removal by the pulmonary filter is slower and therefore delayed in time.
Round 2
Reviewer 2 Report
The authors have diligently addressed all the raised questions and comments, providing detailed explanations. However, it is important to note that despite these responses, the study's limitations remain unchanged. The authors acknowledge these limitations and are aware that they are unable to modify them significantly.
it appears that this second version of the manuscript does not present substantial progress compared to the previous one. Despite the efforts made, the modifications implemented do not seem to have brought significant changes to the article. The gaps or limitations identified in the first version appear to persist, and there are no new pieces of information or analyses that would consider this version as a substantial improvement
Author Response
As authors we take the reviewer’s remarks very seriously. However, the last round of comments is rather confusing. On the one hand the reviewer acknowledges that: first, the publication of these results should be encouraged and second, that we have diligently addressed all the raised questions and comments, providing detailed explanations. On the other hand, the reviewer emphasizes the limitations our study. We totally agree that there are some limitations, and they are well acknowledged thanks to the reviewer suggestions. However, as the reviewer mentioned it is not possible to further modify them significantly. Maybe the reviewer could be more specific about his expectations. We are open to any suggestions.
Since the reviewer refers to the first round of comments, we found one point that have not been addressed which is the hypothesis that the decrease in DLCO could also reduce the amount of nitrogen diffusing into the tissues during body saturation? In short, could altering DLCO also reduce VGE? Therefore, we have added the following sentences:
“We assumed that altered DLCO could slow nitrogen removal and interfere with VGE elimination. However, diffusion is a two-way phenomenon. Movement across selectively permeable membranes are from an area of high concentration to an area of low concentration. Consequently, DLCO could also reduce the amount of nitrogen diffusing into the tissues during body saturation. Although significant inter-personal variability was demonstrated despite all divers following an identical protocol in controlled pool conditions, the higher VGE grade in the DLCO group advocates against this hypothesis.”
“Alternatively, it could be that, in subjects with impaired DLCO, nitrogen removal by the pulmonary filter is slower and therefore delayed in time. Further studies with measurements performed until the 90th minute would be useful to assess this hypothesis.”